# Impaired Kidney Function Associated with Increased Risk of Side Effects in Patients with Small Vessel Vasculitis Treated with Rituximab as an Induction Therapy

**DOI:** 10.3390/jcm10040786

**Published:** 2021-02-16

**Authors:** Aleksandra Rymarz, Anna Matyjek, Magdalena Sułek-Jakóbczyk, Magdalena Mosakowska, Stanisław Niemczyk

**Affiliations:** Department of Internal Diseases, Nephrology and Dialysis, Military Institute of Medicine, 128 Szaserów Street, 04-141 Warsaw, Poland; amatyjek@wim.mil.pl (A.M.); msulek1@wim.mil.pl (M.S.-J.); mmosakowska@wim.mil.pl (M.M.); sniemczyk@wim.mil.pl (S.N.)

**Keywords:** small vessel vasculitis, rituximab, side effects, kidney function, induction therapy

## Abstract

Rituximab (RTX), a monoclonal antibody against the CD20 molecule, is used as an induction therapy in the treatment of small vessel vasculitis (SVV). The aim of the study was to evaluate the efficacy and safety of RTX induction therapy for refractory SVV. A retrospective analysis of 20 patients treated with RTX for active SVV (BVAS/WG ≥ 3) was performed to assess the remission rate and the drug-related severe adverse events 6 months after therapy. The mean age of the studied population was 49 ± 13 years (50% female), 90% of which were PR3-ANCA positive. Complete remission was achieved in 85% of patients, and partial remission was achieved in a further 10% within 6 months after RTX infusions. The remission rate was not influenced by kidney function. Adverse events such as infections (25%), a late onset of neutropenia (10%) and severe hypogammaglobulinemia (5%) were noted. The patients who developed adverse events were older (42 ± 11 vs. 57 ± 12 years; *p* = 0.014) and had a higher serum creatinine level (1.3 mg/dL vs. 3.35 mg/dL; *p* = 0.044). Patients with a glomerular filtration rate (eGFR) lower than 30 mL/min/1.73 m^2^ had a nine-fold higher risk of side effects (OR 9.0, 95%CI: 1.14–71.0). In conclusion, RTX was highly effective as an induction therapy in patients with SVV. Advanced kidney failure with an eGFR lower than 30 mL/min/1.73 m^2^ was one of the risk factors for the occurrence of side effects.

## 1. Introduction

Small vessel vasculitis (SVV) is a disease characterized by the inflammation and necrotizing destruction of the small blood vessel walls, and is predominantly associated with the presence of circulating antineutrophil cytoplasmic antibodies (ANCA). The classification of SVV was established by the International Chapel Hill Consensus Conference (CHCC) in 2012, which divided SVV into ANCA associated vasculitis (AAV) and immune complex SVV [1]. The clinical features distinguish three types of AAV, namely: granulomatosis with polyangiitis (GPA), microscopic polyangiitis (MPA), and eosinophilic GPA (EGPA). However, it has been suggested that the classification should be based on ANCA specificity rather than the clinical features of the disease. Typically, GPA is associated with the presence of anti-proteinase 3 antibodies (PR3-ANCA), and MPA is associated with anti-myeloperoxidase antibodies (MPO-ANCA) [2,3]. Basing a classification on ANCA specificity is supported by many studies that show that relapse rates and clinical outcomes are related to ANCA specificity.

The differences in the clinical course and the outcomes associated with ANCA specificity allow one to expect that the treatment of these diseases may be different. The therapy of SVV consists of two phases. The first is induction therapy and the second is maintenance therapy. Cyclophosphamide (CYC) or rituximab (RTX) together with glucocorticoids (GCS) and/or therapeutic plasma exchanges induce remission. Cyclophosphamide, an old cytotoxic drug, was introduced to the therapy of neoplasms in the 1950s. Its action is associated with the alkylation of guanidine nucleotides, and, as a consequence, does not selectively block cell division [4]. Therefore, it mostly inhibits the rapidly dividing cells. The doses of CYC administrated should be modified according to age and kidney function [5].

Rituximab is a chimeric murine/human monoclonal antibody against the CD20 molecule, which is expressed in the majority of B cells. The depletion of CD20-possitive B cells prevents their transformation to short-lived plasma cells and the production of auto-antibodies. Therefore, the indication for its usage, especially in autoimmunologic disorders, has increased dramatically in recent years. It is used in the treatment of rheumatoid arthritis, systemic lupus erythematosus, and ANCA-associated vasculitis. Aside from its influence on B-cells, recent studies suggest that RTX also changes T-cell actions [6]. An EULAR/ERA-EDTA recommendation published in 2016 suggests using CYC or RTX as a first line induction therapy in organ and life-threatening diseases [7]. However, a RAVE study revealed that rituximab was more effective than cyclophosphamide in PR3-AAV patients. This study showed the remission rate in PR3-AAV patients treated with RTX to be 65% after 6 months compared with 48% after CYC therapy. Therefore, RTX seems to be the preferred option as an induction therapy in these patients [8].

As a maintenance therapy of AAV, azathioprine (AZA), mycophenolate mofetil, or rituximab with concomitant GCS can be used. In the MAINRISTAN study, RTX was more effective than AZA in patients with AAV, irrespective of ANCA type [9].

Among the side effects of RTX treatment, infusion reactions, hypogammaglobulinemia, late onset neutropenia, and infections can be enumerated [2]. The most frequent infectious complications are bacterial infections. They are responsible for 79%of all infections in patients treated with RTX for nephrological indications with pneumonia as a dominant disorder [10]. The infection rate after RTX therapy depends on the disease, which provides an indication for this kind of treatment, and ranges between 1.9 per 100 patient-years for rheumatoid arthritis, and 16.6 per 100 patient-years for lupus erythematous [11,12]. A recently published meta-analysis, which described serious infectious complications after RTX treatment in AAV, reported an incidence rate of 6.5 per 100 patient-years [13]. In this study, the risk factor for the development of these complications was a cumulative dose of RTX. However, this analysis did not consider prognostic factors such as the activity of the disease, degree of kidney failure, or previously used immunosuppressants. In one of the studies conducted on the population of patients with glomerular diseases treated with RTX, the frequency of infections was also related to a cumulative dose of immunosuppression, concomitant diabetes mellitus, or impaired kidney function [10].

In this specific group of patients with SVV, the data, based on clinical studies that considered the side effects of rituximab therapy and the risk factors of their occurrence, such as kidney failure, are extremely limited. Thus, the aim of this study was to evaluate the efficacy and safety of RTX induction therapy used in refractory cases of SVV. This is a group of patients that received immunosuppression and have specific organ damage, including kidney injury. To establish the safety profile of the therapy, we attempted to assess the risk factors of serious adverse events occurring.

## 2. Methods

A retrospective analysis of patients treated with rituximab in the nephrology department between July 2016 and December 2019 was made. All patients received the drug as a second line induction therapy. As a first line induction therapy, intravenous cyclophosphamide was used in all patients. In compliance with the rules of our national healthcare system—which permit the administration of rituximab in cases of inefficiency or side effects of cyclophosphamide—we were able to use this mode of treatment. The activity of the disease should have a Birmingham Vasculitis Activity Score/Wegener Granulomatosis (BVAS/WG) score of at least 3 points in order to fulfill the inclusion criteria [14]. Prior to commencing treatment, in order to exclude active infections—serological markers of hepatitis B and C, QuantiFERON-TB Gold test, urinary and blood culture, C-reactive protein (CRP), and procalcitonin were all verified.

Rituximab was administered four times at weekly intervals, at a dose of 375 mg/m^2^ body surface area. Prior to the administration of the drug, all patients received premedication with an intravenous infusion of methylprednisolone, clemastinum, and paracetamol. For the prevention of infection complications, all patients received trimethoprim-sulfamethoxazole and acyclovir.

The efficacy and safety of the therapy was assessed after 6 months of RTX infusions. The response was defined as follows:complete remission: 0 points in the BVAS/WG score after 6 months;partial remission: drop by 50% in the BVAS/WG score after 6 months;inadequate response: <50% reduction of disease activity in the BVAS/WG score after 6 months.

The adverse events rate over 6 months of follow up was assessed based on medical records. Only serious adverse events were considered, such as infections requiring hospitalization, late onset neutropenia, and severe hypogammaglobulinemia. Late onset neutropenia was defined as neutropenia occurring at least 4 weeks after RTX infusions. Hypogammaglobulinemia was defined as an immunoglobulin G (IgG) level of <400 mg/dL.

Massive hematuria was defined as at least 15 red blood cells (RBC) per high power field (HPF), non-massive hematuria was defined as 5–14 RBC HPF, and no hematuria was defined as <5 RBC HPF. Estimated glomerular filtration rate (eGFR) was calculated according to the short MDRD formula (Modification of Diet in Renal Disease).

The composite outcome was defined as the presence at least one of serious adverse event, such as infections requiring hospitalization, late onset neutropenia, and severe hypogammaglobulinemia. Late onset neutropenia was a neutropenia occurring at least 4 weeks after RTX infusions. Hypogammaglobulinemia was defined as an immunoglobulin G (IgG) level <400 mg/dL. Th study design was accepted by the local ethics committee. All patients signed informed consent for the treatment procedures.

### Statistical Analysis

Continuous variables were described as mean and standard deviation (SD) if normally distributed, or as median with interquartile range (IQR: 25–75% percentile) if distribution was non-normal. Categorical variables were presented as numbers (*n*) with percentages (%). Comparisons of the groups were performed using a t-test for independent samples, as well as the Mann–Whitney test, Fisher’s exact test, and chi-squared test. The results of the baseline assessment and after 6 months from rituximab implementation were compared separately with a paired t-test, Wilcoxon signed-rank test, and chi-squared test. An estimation of the odds ratio (OR) with a 95% confidence interval (95% CI) was used to evaluate the risk of adverse events related to rituximab therapy in advanced chronic kidney disease (CKD) compared with less severe renal injuries. All tests were two-tailed and *p*-values < 0.05 indicated statistical significance.

The statistical analysis was performed using Statistica version 13.1. (TIBCO Software Inc., Palo Alto, CA, USA).

## 3. Results

Twenty patients with SVV were included in the study. The mean age was 49 ± 13 years, and 50% of participants were men. Most of the patients (90%) had the PR3-ANCA phenotype. The median BVAS/WG score was 6 (IQR: 5–9). The clinical characteristics of the study population are presented in Table 1, Table 2 and Appendix A. Among the organs most frequently affected were the lungs and ear–nose–throat (ENT), followed by the kidneys (Table 2).

Eighty-five percent of patients had renal involvement during SVV. Forty-five percent had advanced chronic kidney disease (CKD stage 4–5), and four were treated with dialysis. Among the dialysis patients, three were on hemodialysis therapy and one was on peritoneal dialysis. At the beginning of the induction therapy with RTX, all patients were treated with dialysis for more than 3 months. The main indication for induction therapy with RTX was a flare of SVV, despite the high cumulative dose of CYC (55%). Failure of CYC was defined as the progression of SVV during induction therapy with CYC, which was the reason for the introduction of RTX in 45% of cases. The median cumulative dose of cyclophosphamide before RTX treatment was 152 mg/kg (IQR: 96–303).

### 3.1. Efficacy

Six months after induction therapy with RTX, complete remission was achieved in 85% of patients and partial remission was achieved in a further 10%. One patient (5%) did not achieve any remission (see Figure 1). The remission rate did not differ between patients with an eGFR either lower or higher than 30 mL/min/1.73 m^2^.

Six months after induction therapy with RTX, there was a significant decrease in ANCA titre (*p* = 0.0003), as well as SVV activity, represented by the BVAS/WG score (*p* = 0.0001). Immunoglobulin levels also dropped significantly. However, the percentage of patients with severe hypogammaglobulinemia (<400 mg/dL) did not change significantly (*p* = 0.112). One patient required IgG supplementation before the rituximab treatment because of a low level of IgG (<400 mg/dL), which was followed by another dose before the third infusion of RTX.

The serum creatinine levels and eGFR did not change significantly after 6 months of treatment, whereas the intensity and frequency of hematuria significantly decreased after RTX therapy (*p* = 0.039). The data is presented in Table 3. 

### 3.2. Safety

During the 6 months of follow-up, no patients died. Adverse events were observed in eight patients (40%). These were represented by infections (5 patients—25%), a late onset of neutropenia (2 patients—10%), and severe hypogammaglobulinemia (1 patient—5%). All the patients that suffered from infections had community acquired bacterial pneumonia and one of them had concomitant sepsis. We did not observe opportunistic infections. In five patients with infectious complications, two of them presented stage 5 CKD and three presented stage 4 CKD (Appendix A).

Patients who developed adverse events were older (42 ± 11 vs. 57 ± 12; *p* = 0.014) and had a higher serum creatinine level (1.3 mg/dL vs. 3.35 mg/dL; *p* = 0.044). The data of patients who developed adverse events are presented in Table 4. Patients with advanced CKD, defined as an eGFR lower than 30 mL/min/1.73 m^2^, were more likely to develop adverse events related to RTX therapy (OR 9.0, 95% CI: 1.14–71.0) (Figure 2).

## 4. Discussion

Rituximab, as an immunosuppressive drug, is increasingly used in many renal diseases, including primary and secondary glomerulonephritis, and also in transplant indications. For many years, cyclophosphamide with glucocorticoids have dominated in severe glomerular diseases, including small vessel vasculitis, and still remain an important therapeutical option. However, the serious side effects of CYC and their occasional ineffectiveness necessitates the search for new solutions. The current EULAR/ERA-EDTA recommendation for the management of ANCA-associated vasculitis allows the usage of rituximab as a first line therapy [7]. However, in many countries, rituximab treatment in AAV is still designed solely for refractory or resistant cases, as well as for patients presenting contraindications to CYC. Furthermore, in our analysis, RTX was used in the aforementioned situations. Fifty-five percent of our patients presented a relapse of disease despite receiving a high cumulative dose of CYC and maintenance therapy. CYC was ineffective in 40% of our patients; therefore, it is not surprising that the majority of our patients (90%) presented the PR-3 ANCA phenotype. Detailed analyses have confirmed that SVV with PR3-ANCA antibodies tends to relapse more frequently than MPO-AAV [15,16,17]. In a European Vasculitis Society Trial, PR3-ANCA was an independent risk factor of relapse, with an odds ratio of 1.62 (95% CI: 1.39–1.89) [18]. The traditional classification of SVV based on the clinical course of the disease differentiates GPA, MPA, and EGPA. GPA is strongly associated with the presence of PR-3 ANCA and a predominance of upper respiratory tract and lungs involvement [15]. In accordance with these observations, our study also showed that the most frequently affected organs were ear–nose–throat (75%) and lungs (75%), followed by the kidneys (60%).

In our study, RTX therapy was highly effective. Eighty-five percent of patients reached complete remission, defined as a BVAS/WG score of 0, 6 months after RTX infusions. Regardless of the type of response to RTX, a significant reduction in the BVAS/WG score (*p* = 0.0001) was observed. In 26% of patients, the ANCA level became negative. Other researchers have also reported a high efficiency of this kind of B-cell depletion therapy in SVV. Unizony S et al. observed a significantly better response for RTX than for CYC in patients with PR3-ANCA vasculitis. This group of patients had a twice as great chance of remission after 6 months when treated with RTX rather than CYC (OR 2.11, 95% CI: 1.04 to 4.30). The difference was even more significant for patients treated with RTX because of disease relapse, with an OR of 3.57 (95% CI: 1.43 to 8.93) after 6 months and an OR of 4.32 (95% CI: 1.53 to 12.15) 12 months after RTX infusions [8]. Our observations also followed this trend, as the majority of our patients (90%) presented the PR-3 ANCA phenotype.

The reduction of immunoglobulin levels after B cell depleting therapy is observed frequently and is widely described in the literature [19]. However, a decline in the IgG level is less expressed than an ANCA level decrease [20]. Hypogammaglobulinemia with an IgG level below 400 mg/dL is one of the risk factors for infection complications [20]. Pre-existing hypogammaglobulinemia is associated with a risk of a severe decline in immunoglobulin levels during RTX treatment. Therefore, prior to RTX therapy, immunoglobulin levels should be checked, and in cases of severe hypogammaglobulinemia, supplementation should be introduced [21]. The results of our study also confirmed these observations. A significant reduction in IgG, IgM, and IgA levels was observed 6 months after RTX infusions (*p* = 0.002, 0.0003, and 0.0001). However, the number of patients who developed severe hypogammaglobulinaemia (<400 mg/dL) did not increase significantly (*p* = 0.112).

In our study, in spite of the high rate of a complete remission of SVV, the kidney function did not ameliorate. This could be associated with the fact that RTX was introduced as a second line therapy in the refractory cases, and many of the patients had developed advanced, irreversible kidney damage prior to RTX therapy. In our study population, four patients were treated with renal replacement therapy for more than 3 months. However, a significant reduction in the intensity and frequency of hematuria can be considered a positive result of RTX therapy in kidneys. As was the case in previous studies, the efficacy of RTX was independent of the severity of chronic kidney disease. In our study, the remission rate was the same in patients with an eGFR lower than 30 mL/min/1.73 m^2^ as in individuals with an eGFR above 30 mL/min/1.73 m^2^. Furthermore, Shah S et al. reported a high efficacy of RTX after 6 months in patients with SVV and concomitant severe renal failure, defined as an eGFR lower than 20 mL/min/1.73 m^2^ [22]. However, we observed a higher rate of adverse events in patients with an eGFR below 30 mL/min/1.73 m^2^. This data are consistent with other studies. Trivin et al. reported renal failure as a risk factor for infection events in patients treated with RTX due to various glomerular diseases [13]. However, the threshold of eGFR was at a level of 45 mL/min/1.73 m^2^. Terrier et al. observed a higher rate of infections in patients with non-viral cryoglobulinemia treated with RTX, and concomitant renal failure with an eGFR lower than 60 mL/min/1.73 m^2^ [11]. Scemla et al. reported that renal transplant recipients treated with RTX who developed infectious complications had higher serum creatinine concentrations [23]. In our study, infection complication was observed in 25% of patients, although none of them died. In the study by Terrier et al., the percentage of infectious complications was similar (26%), yet half of them were lethal. According to the literature and recommendations by manufacturers, RTX pharmacokinetics is not dependent on kidney function, and the adjustment of the RTX dose to eGFR is not necessary [24]. However, chronic kidney disease is itself associated with an impaired immune response for pathogens [25]. Moreover, vaccination responsiveness is reduced when compared with individuals without kidney failure [26]. Uremic milieu alters the innate and adaptive immunity through the dysregulation of the pattern recognition receptors, as well as the function of monocytes, neutrophils, and lymphocytes B and T [27]. Therefore, perhaps the therapeutic schedule of RTX treatment for patients with severe renal failure should be adjusted. RTX doses smaller than normal may be effective and safer, but only further studies can resolve this issue.

The limitation of the study is its retrospective design. However, on the basis of the results of this study, a new prospective one may be planned, and the mode of RTX therapy adjusted to kidney function represented by eGFR could be verified. A further limitation of this study is its small sample size and short follow-up—the former is partially associated with the fact that RTX was not introduced as a first line therapy, which limited the number of patients treated. However, the application of RTX therapy only for patients with refractory cases of SVV was as a result of the rules of our national healthcare system, and not a result of a specialist decision. An extension to the length of follow-up is planned, because all of the participants of the study have remained under medical supervision at our center.

In conclusion, RTX was highly effective as an induction therapy in patients with small vessel vasculitis, especially in patients with PR3-ANCA. The remission rate was not related to kidney function. However, advanced kidney failure with an eGFR lower than 30 mL/min/1.73 m^2^, as well as age, were risk factors for the occurrence of side effects. These findings highlight the importance of deeper analyses and further studies considering kidney failure as an important factor in complications assessments.

## Figures and Tables

**Figure 1 jcm-10-00786-f001:**
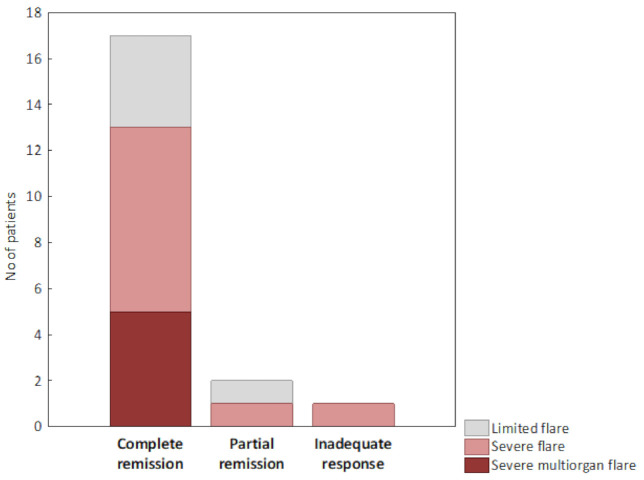
Efficacy of RTX induction therapy.

**Figure 2 jcm-10-00786-f002:**
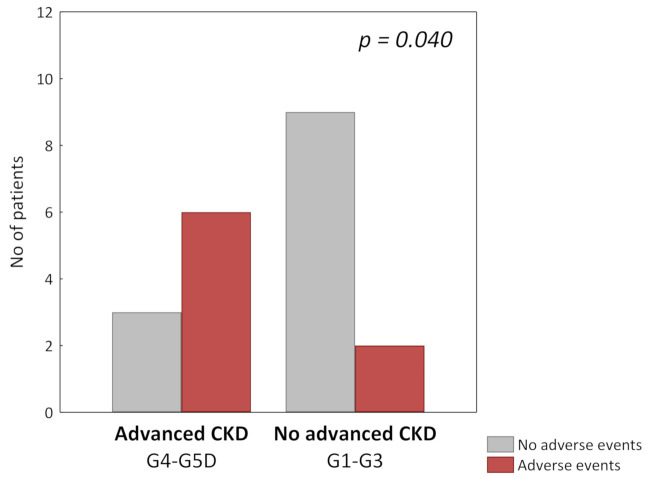
RTX-related adverse events in patients with and without advanced chronic kidney disease (CKD).

**Table 1 jcm-10-00786-t001:** Baseline characteristics of the study population (part 1).

Parameter		RTX Induction *n* = 20
Age (years)	mean ± SD min–max	49 ± 13 24–70
Sex:		
Male	*n* (%)	10 (50%)
Female	*n* (%)	10 (50%)
ANCA type:		
PR3-ANCA	*n* (%)	18 (90%)
MPO-ANCA	*n* (%)	1 (5%)
ANCA negative	*n* (%)	1 (5%)
Duration of SVV (years)	median (IQR) min–max	3 (1–5.5) 0–19
**Treatment of SVV**		
Indication for RTX induction:		
—Failure of CYC	*n* (%)	8 (40%)
—High cumulative dose of CYC	*n* (%)	11 (55%)
—Adverse effects related to CYC	*n* (%)	1 (5%)
CYC cumulative dose (g)	median (IQR) min–max	11 (7.5–23) 0.7–60
CYC cumulative dose:		
>36 g	*n* (%)	3 (15%)
≤36 g	*n* (%)	17 (85%)
CYC cumulative dose (mg/kg)	median (IQR) min–max	152 (96–303) 13–759
CYC cumulative dose:		
>120 mg/kg	*n* (%)	13 (65%)
≤120 mg/kg	*n* (%)	7 (35%)

ANCA—antineutrophil cytoplasmic antibodies; PR3-ANCA—anti-proteinase 3 ANCA; MPO-ANCA—anti-myeloperoxidase ANCA; SVV—small vessels vasculitis; RTX—rituximab; CYC—cyclophosphamide.

**Table 2 jcm-10-00786-t002:** Baseline characteristics of the study population (part 2).

Parameter		RTX Induction *n* = 20
Clinical picture of SVV		
Renal involvement:		
—no ^1^	*n* (%)	3 (15%)
—yes, CKD G1-G3	*n* (%)	8 (40%)
—yes, CKD G4-G5D ^2^	*n* (%)	9 (45%)
Disease activity by BVAS/WG:		
—general symptoms	*n* (%)	9 (45%)
—cutaneous	*n* (%)	5 (25%)
—mucosal membranes/eyes	*n* (%)	1 (5%)
—ear–nose–throat (ENT)	*n* (%)	15 (75%)
—cardiovascular	*n* (%)	0
—gastrointestinal	*n* (%)	1 (5%)
—pulmonary	*n* (%)	15 (75%)
—renal	*n* (%)	12 (60%)
—nervous system	*n* (%)	4 (20%)
—other	*n* (%)	8 (40%)
SVV course:		
—limited flare	*n* (%)	5 (25%)
—severe flare	*n* (%)	10 (50%)
—severe multiorgan flare	*n* (%)	5 (25%)
Dominant organ activity (presence of major symptoms):		
—eye		
—ENT	*n* (%)	1 (5%)
—pulmonary	*n* (%)	4 (20%)
—gastrointestinal	*n* (%)	2 (10%)
—renal	*n* (%)	1 (5%)
—nervous system	*n* (%)	4 (20%)
—other	*n* (%)	3 (15%)
Dominant organ activity (presence of major symptoms):	*n* (%)	4 (20%) *

^1^ eGFR was within the normal range in all of these patients; ^2^ RRT was needed in 44.4% of these patients (4/9 cases). * muscular involvement in 1 patient, inflammatory aortic disease in 3 patients (of the ascending aorta in 1 case and of the abdominal area in 2 cases).

**Table 3 jcm-10-00786-t003:** Comparison of patients’ status at baseline and 6 months after RTX.

Parameter		Baseline	Month 6	*p*-Value
BVAS/WG score	median (IQR) min–max	6 (5–9) 3–12	0 (0–0) 0–4	0.0001 *
ANCA titre (IU/mL)	median (IQR) min–max	30 (10–124) 0–177	6.2 (0–24) 0–174	0.0003 *
IgA (mg/dL)	mean ± SD min–max	196.5 ± 66.7 109–330	141.9 ± 53.2 58–247	0.0001
IgM (mg/dL)	median (IQR) min–max	65 (34–117) 26–164	39.5 (28–65) 10–127	0.0003 *
IgG (mg/dL)	mean ± SD min–max	969.5 ± 320.7 369–1740	726.3 ± 243.3 325–1230	0.002
IgG (mg/dL) <400 400–699 ≥700	*n* (%) *n* (%) *n* (%)	1 (5%) 3 (15%) 16 (80%)	1 (5%) 9 (45%) 10 (50%)	0.112 ^#^
Serum creatinine (mg/dL)	median (IQR) min–max	1.6 (0.9–2.5) 0.8–4.5	1.4 (1.0–2.1) 0.8–5.7	1.000
eGFR ^ (mL/min/1.73 m^2^)	median (IQR) min–max	54 (25–79) 15–108	55 (29–75) 11–90	0.691 *
Hematuria ^				0.039
—yes, massive	*n* (%)	5 (31.25%)	0	
—yes, non-massive	*n* (%)	5 (31.25%)	5 (31.25%)	
—no	*n* (%)	6 (37.5%)	11 (68.75%)	
Proteinuria (mg/dL)	median (IQR)	29 (10–46)	21 (10–43)	0.722

* Wilcoxon’s signed-rank test; ^#^ Pearson’s chi-squared test; ^ comparison performed on non-dialyzed patients only, dialyzed cases were excluded from analysis.

**Table 4 jcm-10-00786-t004:** Comparison of patients with and without adverse events related to RTX therapy.

Parameter		No Adverse Events *n* = 12	Adverse Events *n* = 8	*p*-Value
Age (years)	mean ± SD	42 ± 11	57 ± 12	0.014 *
Sex: Male Female	*n* (%) *n* (%)	5 (41.7%) 7 (58.3%)	5 (62.5%) 3 (37.5%)	0.650
Duration of AAV (years)	median (IQR)	3 (3–10.5)	1.5 (1–4)	0.107
CYC cumulative dose (g)	median (IQR)	18 (9.63–35.5)	8.75 (6.5–10.38)	0.082
CYC cumulative dose: >36 g ≤36 g	*n* (%) *n* (%)	3 (25%) 9 (75%)	0 8 (100%)	0.242
CYC cumulative dose (mg/kg)	median (IQR)	185 (133–465)	105 (82–209)	0.153
CYC cumulative dose: >120 mg/kg ≤120 mg/kg	*n* (%) *n* (%)	10 (83.3%) 16.7%)	3 (37.5%) 5 (62.5%)	0.062
RTX single dose (mg)	mean ± SD	721 ± 96	700 ± 122	0.682
Baseline BVAS/WG score	median (IQR)	5 (4.5–8)	6 (6–10)	0.111
Baseline ANCA titre (IU/mL)	median (IQR)	37 (4–127)	26 (8–73)	0.847
Serum creatinine (mg/dL)	median (IQR)	1.3 (0.85–2.05)	3.35 (2.2–4.9)	0.044
eGFR (mL/min/1.73 m^2^) G1-G3: ≥30 G4-G5D: <30	*n* (%) *n* (%)	9 (75%) 3 (25%)	2 (25%) 6 (75%)	0.040 ^#^
eGFR (mL/min/1.73 m^2^)	mean ± SD	57 ± 32	29 ± 26	0.049
IgA (mg/dL)	mean ± SD	207.2 ± 65.5	175.9 ± 71.4	0.337 *
IgM (mg/dL)	median (IQR)	67 (53.5–128.5)	52.5 (31.5–85)	0.203
IgG (mg/dL)	mean ± SD	995.7 ± 317.3	921.9 ± 345.8	0.636 *

*p*-values for Mann–Whitney’s test, * *p*-value for t-test, ^#^*p*-value for chi-squared test.

## Data Availability

The data presented in this study are available on request from the corresponding author.

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
