# Peer review of "Impaired Kidney Function Associated with Increased Risk of Side Effects in Patients with Small Vessel Vasculitis Treated with Rituximab as an Induction Therapy"

_jcm, 2021, doi:10.3390/jcm10040786_

Round 1

Reviewer 1 Report

I was a reviewer for this manuscript before. I think the investigators have significantly improved their manuscript.

Quality of manuscript presentations can be better. Figure colors can be in color mode given this is open access online.

Introduction and discussion can also be improved. 

Furthermore, what is the future/next step study that should be conducted? can be adding on after limitations of the study. 

Author Response

Thank You for the revision and comments. English style has been corrected.

Does the introduction provide sufficient background and include all relevant references?     

The introduction was enriched including the new references.

Quality of manuscript presentations can be better. Figure colors can be in color mode given this is open access online.

The figures have been modified and their colours were changed.

Introduction and discussion can also be improved. 

The introduction and discussion has been modified. The modified parts have been highlightened in green colour.

Furthermore, what is the future/next step study that should be conducted? can be adding on after limitations of the study. 

The limitations of the study have been enumerated at the end of the discussion with the potential future steps for continuation of the study. (lines: 291 – 299).

Reviewer 2 Report

Rymarz et al., report results of a cohort study of 20 patients with small vessel vasculitis treated with rituximab. They find a high level of efficacy in controlling disease activity but also observed 25% had serious infections requiring hospitalization after treatment (however, these look comparable to ritxiumab immunosuppressant-related serious infections in Lupus and other diseases). I have some questions for the authors to improve the presentation—my main concern is adding more description to how adverse events were measured.

  1. Suggest making it clear in the introduction what is new/novel about this study. Is this the first study in SVV? But others have reported in ANCA specifically? If so, it still seems like most were ANCA associated in this study. If there aren’t major novel aspects of this study, suggest just making this clear in the intro, describing this as a replication study.
  2. In the methods, suggest defining what the BVAS/WG score is. I realize most folks working in vasculitis will understand “remission” definitions but this should be included for completeness.
  3. More information is needed on how adverse event data were collected. Was this chart review? Questionnaire? Were these reviewed by a physician? Were they classified as treatment associated? In a prospective clinical trial there is a lot more structure and standardization in how these would be gathered, and I’m worried a reader might interpret these adverse event as if they were from a trial. I think it might be easier and more accurate to describe specifically which events you considered serious adverse events and how you looked for those in the chart (assuming it was chart review).
  4. Related to above—I see patients signed informed consent for treatment procedures. Assuming that means everything was prospectively collected? If so, I would describe this as a prospective—not a retrospective study throughout.
  5. Was proteinuria measured as a spot urine sample? Were these first morning voids? Suggest reporting protein: creatinine ratio instead. Or is this some kind of dipstick test rest?
  6. Can you provide more information on the types of infections?
  7. Do you have any information on the initial adverse response rate to CYC? How many had serious infection after induction CYC dose? Does initial CYC tolerance predict decreased likelihood of infection from rituximab?
  8. I appreciate the strong efficacy findings, but 1 in 4 being hospitalized for severe infection seems serious. I’m not deeply familiar with SVV, but suggest you make a stronger case for why it would be “worth it”. What are the consequences of active SVV/not in remission?

Author Response

Thank You for the revision and comments. English style has been corrected.

Does the introduction provide sufficient background and include all relevant references?     

The introduction was enriched including the new references.

Are the methods adequately described?

The method section was improved.

Comments and Suggestions for Authors

Rymarz et al., report results of a cohort study of 20 patients with small vessel vasculitis treated with rituximab. They find a high level of efficacy in controlling disease activity but also observed 25% had serious infections requiring hospitalization after treatment (however, these look comparable to ritxiumab immunosuppressant-related serious infections in Lupus and other diseases). I have some questions for the authors to improve the presentation—my main concern is adding more description to how adverse events were measured.

  1. Suggest making it clear in the introduction what is new/novel about this study. Is this the first study in SVV? But others have reported in ANCA specifically? If so, it still seems like most were ANCA associated in this study. If there aren’t major novel aspects of this study, suggest just making this clear in the intro, describing this as a replication study.

The introduction has been modified to explicitly emphasize kidney failure as a risk factor resulting in the occurrence of adverse events in small vessel vasculitis patients (lines: 97 – 99).

  1. In the methods, suggest defining what the BVAS/WG score is. I realize most folks working in vasculitis will understand “remission” definitions but this should be included for completeness.

An explanation of BVAS/WG score has been added in the Methods section with an appropriate reference 13 – in the reference list:

Stone JH, Hoffman GS, Merkel PA, Min YI, Uhlfelder ML, Hellmann DB, Specks U, Allen NB, Davis JC, Spiera RF, Calabrese LH, Wigley FM, Maiden N, Valente RM, Niles JL, Fye KH, McCune JW, St Clair EW, Luqmani RA; International Network for the Study of the Systemic Vasculitides (INSSYS). A disease-specific activity index for Wegener's granulomatosis: modification of the Birmingham Vasculitis Activity Score. International Network for the Study of the Systemic Vasculitides (INSSYS). Arthritis Rheum. 2001 Apr;44(4):912-20. doi: 10.1002/1529-0131(200104)44:4<912::AID-ANR148>3.0.CO;2-5. PMID: 11318006.

  1. More information is needed on how adverse event data were collected. Was this chart review? Questionnaire? Were these reviewed by a physician? Were they classified as treatment associated? In a prospective clinical trial there is a lot more structure and standardization in how these would be gathered, and I’m worried a reader might interpret these adverse event as if they were from a trial. I think it might be easier and more accurate to describe specifically which events you considered serious adverse events and how you looked for those in the chart (assuming it was chart review).

The study was retrospective not prospective. The patients were treated and observed in one specialized center during the entire period of follow-up. Therefore, all adverse events were confirmed in this center. The method section has been modified to avoid the interpretation that our study is a prospective trial.

  1. Related to above—I see patients signed informed consent for treatment procedures. Assuming that means everything was prospectively collected? If so, I would describe this as a prospective—not a retrospective study throughout.

In our center we have a schedule of RTX treatment with a list of diagnostic tests undertaken periodically after RTX treatment. For all of these procedures (therapy and the monitoring), the patient has to sign their informed consent. However, the study was not planned before the beginning of these therapies, therefore it is not prospective.

 Was proteinuria measured as a spot urine sample? Were these first morning voids? Suggest reporting protein: creatinine ratio instead. Or is this some kind of dipstick test rest?

It was a first morning spot urine sample. The protein: creatinine ratio is not available.

  1. Can you provide more information on the types of infections?

It has been added to the Result section:

“All patients that suffered from infections had bacterial pneumonia and one of them had concomitant sepsis.” (lines: 198 – 199).

  1. Do you have any information on the initial adverse response rate to CYC? How many had serious infection after induction CYC dose? Does initial CYC tolerance predict decreased likelihood of infection from rituximab?

In our study neither the cumulative dose of cyclophosphamide which was received before RTX treatment nor the cumulative dose per kg of body weight was a risk factor of serious adverse events. Infectious complications after the induction of CYC were mostly unavailable because many patients had a long history of diseases (for example 10 years) and were initially treated in other centers.

  1. I appreciate the strong efficacy findings, but 1 in 4 being hospitalized for severe infection seems serious. I’m not deeply familiar with SVV, but suggest you make a stronger case for why it would be “worth it”. What are the consequences of active SVV/not in remission?

As we presented in the Result section all patients in our study with the infectious complications recovered. No patients died. To compare, in the study made by Trivin et al. [12] half of the patients with infectious complications died. The consequences of active SVV not being in remission in patients with severe disease (BVAS/WG at least 3 points) are extremely serious including death or end stage insufficiency of many organs: lungs, kidneys, central nervous system. Therefore despite the 25% of severe adverse event it is necessary to treat these patients.

This manuscript is a resubmission of an earlier submission. The following is a list of the peer review reports and author responses from that submission.

Round 1

Reviewer 1 Report

Overall, this study has major limitations. The investigators claimed Impaired kidney function was associated with increased risk of side effects in patients with small vessel vasculitis treated with rituximab as an induction therapy.

However, it is unclear what is definition of "adverse effects" mean in the method. In addition, the investigators did not described the details about percentage of each adverse effect in the result. 

These adverse effect, for example, "nausea/vomiting" could be due to the low GFR itself, not because of medication/Rituximab. It is already known that those with low eGFR would have these symptoms, but these are not because of medication.

Data on CD19-CD20, and also proteinuria are lacking.

Data on measure GFR or creatinine clearance are lacking

Reviewer 2 Report

Rymarz et al. reported a possible association between CKD and increased risk of side effects in patients treated with rituximab for small vessel ANCA vasculitis. 

Baseline eGFR may be a potential prognostic factor for predicting side effects. However, the relationship between eGFR and side effects may be confounded by dose of steroid before the rituximab therapy or treatment failure due to inadequate dose of cyclophosphamide. The same is true for the factors such as baseline IgM/IgG.
Furthermore, authors defined side effects as a composite outcome including infusion reactions, infections requiring hospitalization, late onset neutropenia, and severe hypogammaglobulinemia defined as immunoglobulin G (IgG) level <400 mg/dL). Although the frequency of hard outcomes such as infections requiring hospitalization may not be so high and the sample size of this study may be small for detecting the difference in hard outcomes, clinician's primary interest should be infections requiring hospitalization. Therefore, small sample size is a large limitation of this study.